# Silver nanoparticle biosynthesis utilizing *Ocimum kilimandscharicum* leaf extract and assessment of its antibacterial activity against certain chosen bacteria

**Horyomba Siaka Ouandaogo**[1]*, **Souleymane Diallo**[2], **Eddy Odari**[3], **Johnson Kinyu**[4]

**1** Department of Molecular Biology and Biotechnology, Pan African University of Basic Science, Technology and Innovation, Juja, Kenya, **2** International Centre of Insect Physiology and Ecology, Nairobi, Kenya, **3** Department of Medical Microbiology, Jomo Kenyatta University of Agriculture and Technology, Juja, Kenya, **4** Department of Biochemistry, Jomo Kenyatta University of Agriculture and Technology, Juja, Kenya

* siakaouandaogo@gmail.com

## Abstract

The use of plants in the biological production of silver nanoparticles for antibacterial applications is a growing field of research. In the current work, we formulated *Ocimum kilimandscharicum* extracts using silver nanoparticles, and evaluated its potential antibacterial activity. Aqueous and methanol plant extracts were used to reduce silver nitrate at different time intervals (30 to 150 minutes) and pH (2 to 11). The UV-visible absorption spectrum recorded for methanol and aqueous extracts revealed a successful synthesis of AgNPs for methanol and aqueous extracts. The antimicrobial activity of the AgNPs was evaluated against *Escherichia coli* ATCC 25922, *Salmonella choleraesuius* ATCC 10708, and *Staphylococcus aureus* ATCC 25923 The best inhibition zone for the methanol and aqueous-mediated AgNPs, ranging from 12 ± 1 to 16 ± 1mm. Additionally, the methanol and aqueous extract silver nanoparticles had the same Minimum Inhibitory Concentration (6.25 ± 0.00 mg/ml), whereas the Minimum Bactericidal Concentrations were 12.5 ± 0.00 and 25 ± 0.00 mg/ml, respectively. The highest inhibition zone of 16 ± 1 mm was observed against *Salmonella choleraesuius* with 50 ± 0.00 mg/ml aqueous silver nanoparticles. The results show that the silver nanoparticles made with *Ocimum kilimandscharicum* have antibacterial action against those microorganisms.

**Data Availability Statement:** All relevant data are within the manuscript and its Supporting Information files. Like The values behind the

## Introduction

The genus Ocimum, renowned globally among the numerous traditionally significant plants, has been utilized for thousands of years due to its diverse medicinal properties [1].

A native of East Africa, *Ocimum kilimandscharicum* is a member of the Lamiaceae family and is currently planted worldwide. This *Ocimum* species has a harsh, unpleasant flavor. This plant is around 2.44 meters in height and can easily be recognized by its shrubby form and pubescent understory shrub with quadrangular branchlets [2]. The leaves are generally ovate

means, standard deviations and other measures reported; The files (others) I uploaded contain: - The values used to build graphs; - The points extracted from images for analysis. For clarification: OK: O. kilimandscharicum; AES: Aqueous Extract Silver; MES: Methanolic Extract Silver

**Funding:** Horyomba Siaka Ouandaogo MB401-0005/2019 AFRICAN UNION https://au.int/ No: The funders had no role in study design, data collection and analysis, decision to publish, or preparation of the manuscript.

**Competing interests:** The authors have declared that no competing interests exist.

shaped with pale-yellow blooms [3]. Commonly known as Camphor basil, plant gained popularity in cosmetics and pharmacological properties. Indeed, *Ocimum kilimandscharicum*, is frequently used in traditional medicine to cure a broad range of many conditions, including diarrhea of many illnesses, including diarrhea, measles, stomach discomfort, and colds and coughs [3,4]. Additionally, the plant extracts have been found to have wound-healing properties [5]. In pest control, the extracts with biologically active components contain insecticidal, fungistatic [6]. They are often used to enhance drug delivery systems and reduce the toxicity of many candidates active phytocompounds. Nanotechnology applications in agriculture, such as crop protection, controlled pesticide release, target gene transfer, plant hormone administration, and the use of nano sensors for early disease detection in plants, are expanding rapidly [7–9]. In phytomedicine, Phyto drugs are encapsulated in a drug carrier that facilitates their absorption and consequently delivered into the bloodstream [10].

Due to their distinct physiochemical features, silver nanoparticles are well recognized for displaying high antimicrobial action against bacteria [11], fungi [12] and viruses [13]. A wide variety of human infections have been targeted through the extensive use of silver [14]. Many researchers have shown that silver nanoparticles are effective against various plant infections because they are antibacterial and antifungal [15]. Thus, various nanocarriers have been developed to facilitate drug absorption and toxicity. However, their efficacy and safety are highly dependent to nanomaterial used.

For example, metallic nanoparticle manufacturing via chemical and physical means is frequently expensive and involves potentially harmful substances. Numerous biological methods have been created as an alternative to create safe, affordable, and environmentally acceptable nanoparticles [16]. It is anticipated that employing plants for biological synthesis would offer several advantages over bacteria and fungi. Two of these advantages are the accessibility of plants and the existence of a wide variety of metabolites that help produce silver nanoparticles.

The use of *O. kilimandscharicum* in the manufacture of AgNPs was not reported in the literature. Herein, we aimed to bio-synthesize AgNPs using *O. kilimandscharicum* and assess its efficacy in managing bacterial infections.

## 2. Materials and methods

### 2.1. Chemicals

Legacy Lab Africa Ltd., a Kenyan company, provided all the chemicals used in this investigation. Analytical-grade chemicals were employed throughout.

### 2.2. Preparation of plant extract

*Ocimum kilimandscharicum* leaves were collected from the Juja region in Kenya in March 2021 and transported to the Pan African University of Basic Sciences, Technology and Innovation, Kenya (PAUSTI). Mr John Kamau Muchuku, Department of Botany, Jomo Kenyatta University of Agriculture and Technology, identified and verified the plant leaf material, and a voucher specimen (HSO–JKUATBH/001/A– 2021) was put in the herbarium for further referencing.

Fresh leaves were, surface-cleaned with running water, and rinsed with distilled water. The leaves were then chopped into little pieces, allowed to air dry at room temperature in the shade and ground into a fine powder with an electric blender. Following extraction with 2000 ml of methanol (70%), 100 g of the finely powdered leaf extract was placed in an orbital shaker incubator at 25˚C for 72 hours. A rotatory evaporator was used to concentrate the methanolic extract filtrate. 100 g of the finely powdered leaf extract was placed in 2000 ml of distilled water

at 80˚C. After 1 hour, the mixture was filtrated using Whatman filter paper [17]. The resultant extracts were stored in a refrigerator at 4˚C.

## 2.3. Green synthesis of silver nanoparticles

**2.3.1. Preparation of the 1 mM AgNO₃ solution.** We utilized the approach described by Ghabban *et al.* [18] with minor adjustments. Briefly, 1.7 g of silver nitrate was dissolved in 1000 ml of deionized water in a 1000 ml volumetric flask and then transferred into an amber bottle to create a stock solution of 10 mM of AgNO3 aqueous solution.

From the stock, 1 mM of AgNO3 aqueous working solution was created by measuring 10 ml of 10 mM AgNO3 and adding 90 ml of distilled water. The flask was wrapped in aluminum foil to avoid photoreduction and kept in the dark.

**2.3.2. Synthesis of silver nanoparticles.** In a conical flask, three milliliters of both methanolic and aqueous plant extracts were combined with one hundred milliliters of a 1 mM silver nitrate solution. The bio reduction process (Ag+→Ag0) was initiated by covering the mixture with aluminum paper to shield it from sunlight and heating the mixture to 80˚C. After 90 minutes, the formation of silver nanoparticles was confirmed by observing a color change from colorless (AgNO3) to light brown to dark brown (AgNO3+extract solution)[19]. UV-vis spectrophotometer absorbance readings across the range of 200 nm to 800 nm further verified the presence of silver nanoparticles (AgNPs) in the reaction mixture. To eliminate impurities, the solution underwent centrifugation at 18,500 rpm for 60 minutes at 4˚C and was subsequently freeze-dried. Two rounds of centrifugation were employed for purification purposes [19]. Fig 1 displays the *O. kilimandscharicum* plant and the synthesized nanoparticles.

## 2.4. Synthesized silver nanoparticles characterization

**2.4.1. Analysis of UV-Visible spectroscopy.** PAUSTI's UV-Visible spectrophotometer (Model 6800, Jenway) was used to evaluate the reduction of silver nitrate ions (Ag+) by *Ocimum kilimandscharicum* leaf extract at wavelengths between 800 and 300 nm [20,21]. According to Singh *et al.* [14], and Shaik *et al.* [22] phenolics, flavonoids, and tannins play a vital role during redox reaction and act as capping agents. The 400–500 nm peak, a typical peak of silver nanoparticles owing to surface resonance plasmon (SRP), was used to validate the synthesis of silver nanoparticles.

**2.4.2. Analysis of functional group using FTIR.** Fourier infrared spectrometry was used to classify functional groupings (FTIR, Shimadzu 8400, Kyoto, Japan). We combined 10 mg of powdered AgNPs with 100 mg of Potassium Bromide to make the salt disc. A range of 4000 to 500 cm-1 was used to capture the spectra [21].

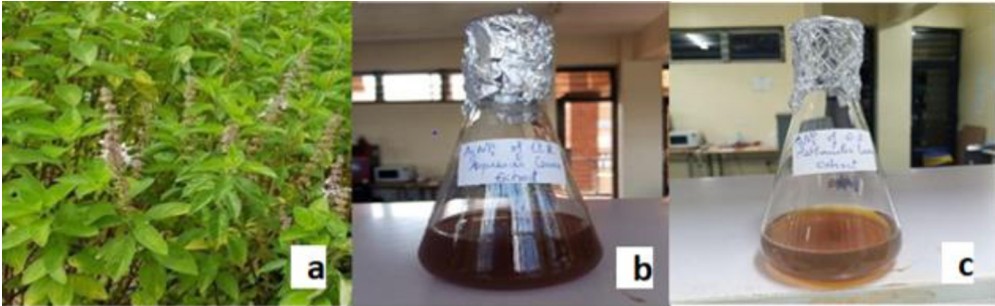

**Fig 1. a,** *O. kilimandscharicum* plant; **b,** synthesized AgNPs using the aqueous extract; **c,** synthesized AgNPs using the methanolic extract.

**2.4.3. Dynamic light scattering.**   Malvern ZETASIZER NANO was utilized to determine the particle size, polydispersity index (PDI), and stability of biosynthesized AgNPs (Model: ZEN5600, Serial number: MAL1168679) [23].

## 2.5. Antimicrobial screening of the extracts and their silver nanoparticles

**2.5.1. Organisms testing.**   *O. kilimandscharicum* leaf extracts and their silver nanoparticles were examined for their antibacterial activity against Gram-positive and negative bacteria strains. All strains are American Type Culture Collection (ATCC). The Kenya Medical Research Institute (KEMRI) donated the bacterial strains, *E. coli (ATCC 25922), S. choleraesuius (ATCC 10708*), and *S. aureus (ATCC 25923*).

**2.5.2. Bacterial cultures, upkeeping, and preparation of the inoculum.**   The selected bacteria were subcultured and kept on nutrient agar. The bacterial strains were cultured overnight in Muller Hinton broth (MHB) at a concentration of 0.5 McFarland standard.

**2.5.3. Zone of inhibition (ZOI).**   The antibacterial activity of the crude extract and silver nanoparticles was assessed using the Agar diffusion technique, as described by Ali *et al.*, Ahmad *et al.*, Shanmugapriya *et al.* [24–26] with slight modifications. 5 Muller Hinton Agar (5.6 grams) was dissolved in 200 milliliters of distilled water, heated, and autoclaved under specific parameters. Subsequently, 20 ml of the prepared media was poured into each petri dish and allowed to solidify. Newly subcultured bacteria were transferred to the plates using sterile swabs. Using a sterile cork borer, six (6) mm wells were created on each plate. In summary, 100 μl of the prepared extract solution (50 mg/ml) and (50 mg/ml) of silver nanoparticles were pipetted into the wells and allowed to diffuse. Gentamycin (10 mg) served as the positive control. After 24 hours of incubation, the inhibition zones were measured in millimeters.

**2.5.4. Determination of the minimum inhibitory concentration (MIC).**   The minimum inhibitory concentration of O. kilimandscharicum extract and the corresponding silver nanoparticles was determined in triplicates using the resazurin-based 96-well plate microdilution method [27]. Doses ranging from 200 mg/ml to 0.39 mg/ml were prepared through two-fold dilution. Each well received 200 μl of newly cultivated bacteria and 200 μl of the test samples. The plates were then incubated for 24 hours at 37°C. On the following day, 100 μl of resazurin dye was added to each well and allowed to incubate for an additional two hours. The minimum inhibitory concentrations were identified by observing the color changes in each well, transitioning from blue to pink, indicating the lowest inhibitory concentrations [28].

**2.5.5. Determination of the minimum bactericidal concentration (MBC).**   The minimum bactericidal concentration (MBC) is defined as the lowest concentration at which colony growth is inhibited. In line with the methodology proposed by Lemma *et al*. and Mkangara *et al*. [29,30] with some modifications, extracts of O. kilimandscharicum and silver nanoparticles were assessed to determine their bactericidal concentrations. A sterile swab was used to transfer the contents from wells (3 wells downstream) where no color changes had occurred to fresh sterile Petri plates for further incubation. After an additional twenty-four hours of culture, the plates were examined for bacterial growth. The minimum bactericidal concentration was extrapolated from the 96 plates to those without observable growth.

## 2.6. Statistical analysis

The Mean ± standard error (SE) was used to express the experimental results. R studio, Excel, and SAS9.2 were used to perform an analysis of variance (ANOVA). The significant difference between the means was performed using the Turkey test. P values less than 0.05 were considered significant.

## 3. Results and discussion

### 3.1. Silver synthesis and characterization

There are several ways to synthesize nanoparticles, including chemical, physical, and biological approaches. Although physical and chemical techniques are the most popular, they are both damaging to the environment and costly [31]. Plants are abundantly accessible, inexpensive, and contain bioactive substances [32]; hence, green nanoparticle manufacturing has lately attracted considerable attention. Moreover, using plants to synthesize is safe and ecologically beneficial [14,33]. Plants are rich in secondary metabolites, which usually facilitate the synthesis of silver nanoparticles. In a research study conducted by Saxena *et al.* [34], flavonoids, alkaloids, glycosides, saponins, and tannins were present in the methanolic extract of *O. kilimandscharicum*.

**3.1.1. UV-Visible spectroscopy.** An aliquot of the solution was analyzed using UV-visible spectrophotometry to track the development of silver nitrate bioreduction. Figs 2 and 3 distinctive display peaks measured at 424 nm for the methanolic extract and 442 nm for the aqueous extract [35,36], respectively.

*Produced AgNP's stability over time*. AgNP suspension was subjected to different time conditions, and the absorbance spectra of UV/Vis were in the scan range of 350nm to 650nm. The times tested were 30 mn, 60 mn, 90 mn, 120 mn, and 150 mn (Figs 4 and 5).

Ninety minutes (90 min) is the best time to stop our synthesis process for the AgNPs using *O. kilimandscharicum* methanolic leaves extract, corresponding to a sharp peak (Fig 4). In comparison, 120 minutes is the best time for *O. kilimandscharicum* aqueous extract mediated AgNPs (Fig 5). The difference in optimal synthesis times for silver nanoparticles (AgNPs) using *O. kilimandscharicum* methanolic and aqueous extracts can be attributed to variations in the chemical composition and properties of the extracts. Methanolic extracts typically contain a higher concentration of phytochemicals such as polyphenols, tannins, and terpenoids [1], which may facilitate a faster reduction of silver ions and promote nanoparticle formation. This accelerated reduction process could result in the earlier attainment of optimal synthesis conditions, evidenced by the distinct peak observed at ninety minutes (90 min) in Fig 4. Conversely, aqueous extracts may have a different composition and concentration of bioactive compounds compared to methanolic extracts. This distinction can affect the kinetics of the reduction

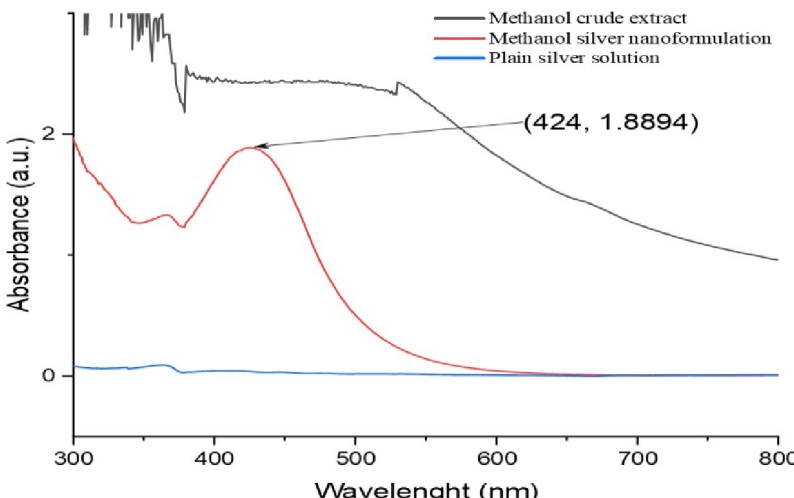

**Fig 2. UV/Vis spectra of *O. kilimandscharicum* methanolic extract mediated AgNPs.**

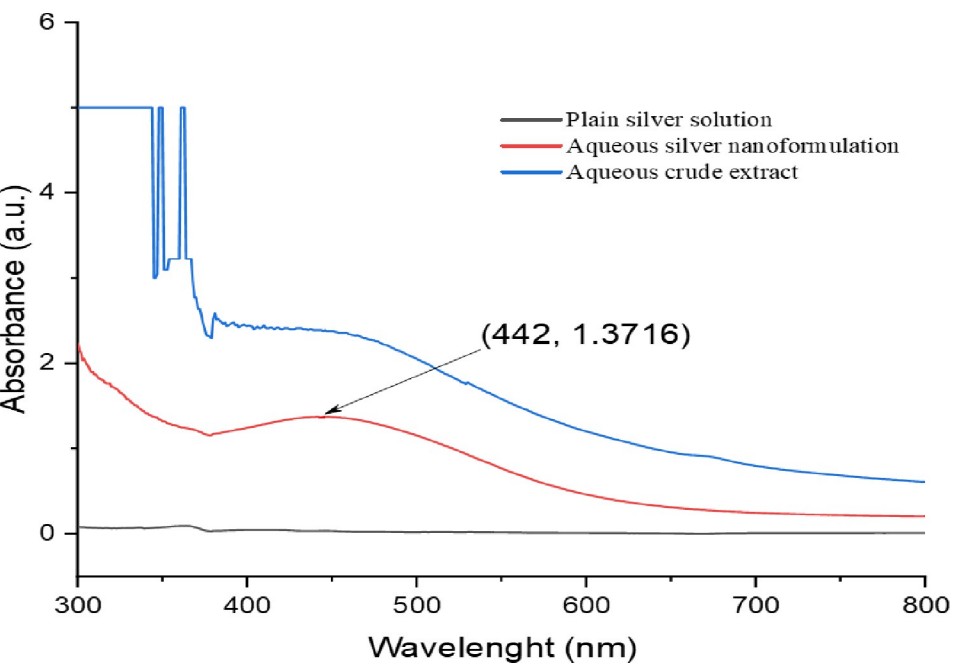

**Fig 3. UV/Vis spectra of *O. kilimandscharicum* aqueous extract mediated AgNPs.**

reaction and the nucleation and growth of nanoparticles. In the case of AgNPs synthesized with *O. kilimandscharicum* aqueous extract, the slower reduction kinetics or lower concentration of reducing agents may necessitate a longer synthesis time for achieving optimal nanoparticle synthesis conditions. This longer duration allows for sufficient reduction of silver ions

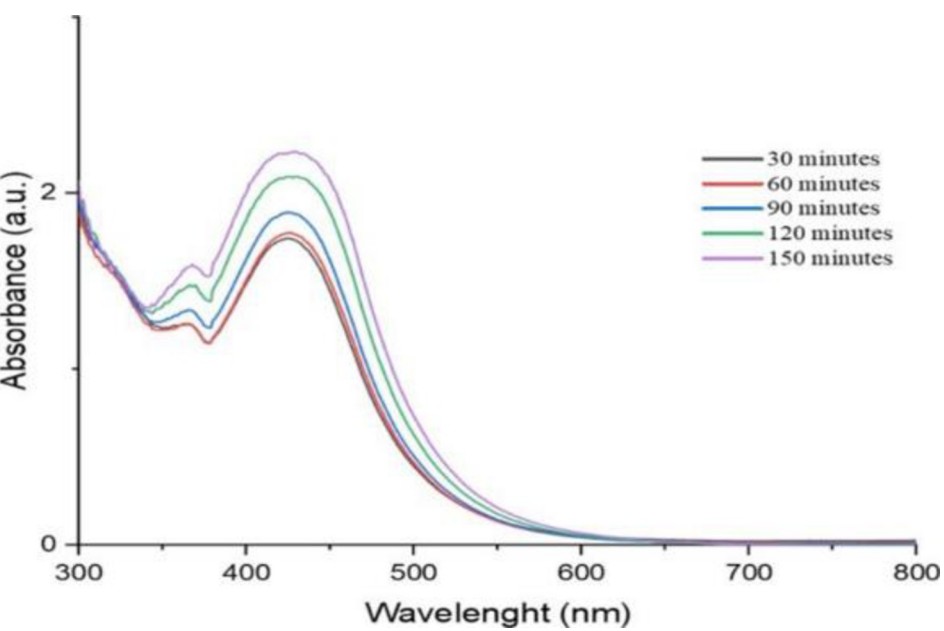

**Fig 4. UV/Vis spectra of time stability of *O. kilimandscharicum* methanolic extract mediated AgNPs.**

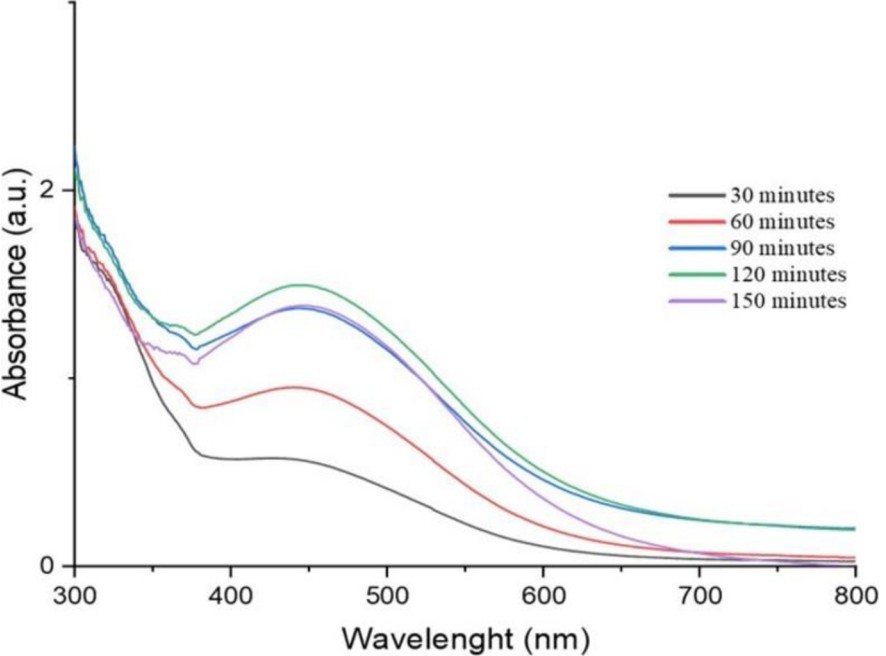

**Fig 5. UV/Vis spectra of time stability of *O. kilimandscharicum* aqueous extract-mediated AgNPs.**

and the formation of nanoparticles, resulting in the prominent peak observed at 120 minutes in Fig 5.

*Produced AgNP's stability pH.* The produced AgNP suspension was aliquoted into 5 test tubes, each holding about 3 ml of the AgNP suspension. The suspensions in the test tubes were adjusted to various pH settings ranging from 2 to 11 using drops of either 1N NaOH or 1N HCl. The absorbance was measured from 300 to 800 nm (Figs 6 and 7).

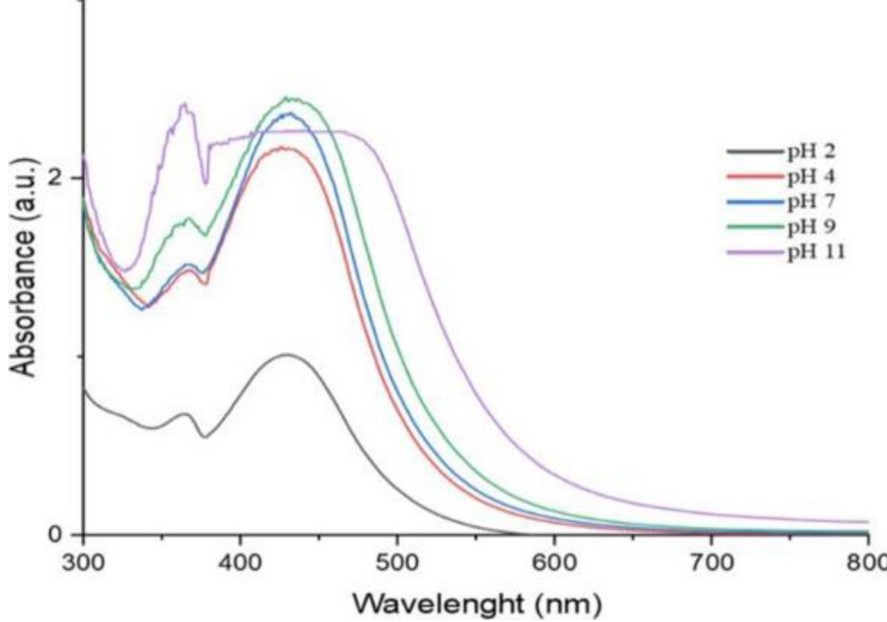

**Fig 6. UV/Vis spectra of Ph stability of *O. kilimandscharicum* methanolic extract mediated AgNPs.**

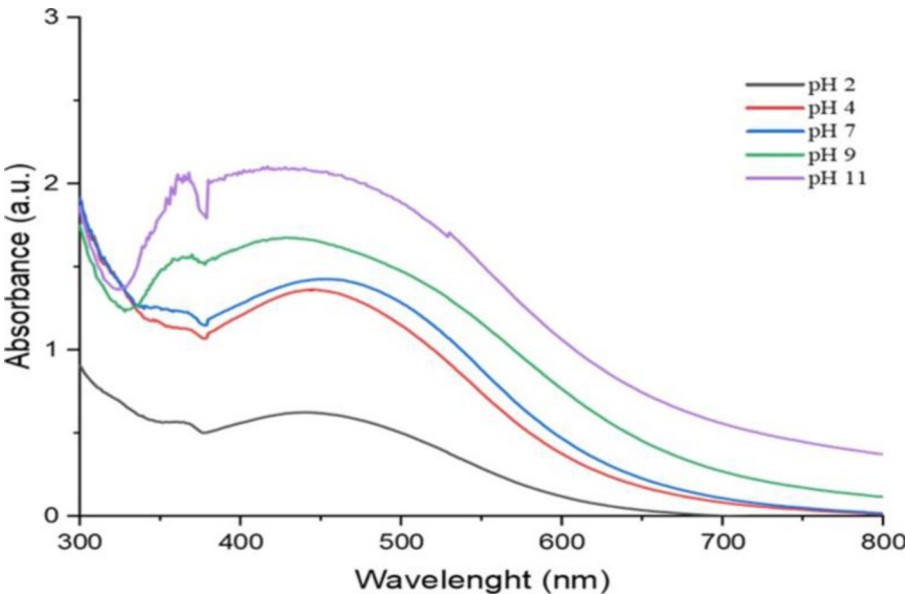

**Fig 7. UV/Vis spectra of Ph stability of *O. kilimandscharicum* aqueous extract mediated AgNPs.**

Both AgNPs synthesized using the methanolic and aqueous extracts are good at pH 2, 4, and 7, showing that the AgNPs are stable at these pHs (Figs 6 and 7). Above pH 7, the AgNPs are no longer stable, as revealed by the very broad peaks (Figs 6 and 7). Silver nanoparticles have a tendency to agglomerate due to their high surface energy. To counteract this, various substances such as plant extracts, gums, cationic surfactants, and polymers have been employed to stabilize colloidal metal dispersions. These capping agents play a crucial role in preventing the agglomeration of colloidal particles by enveloping them. The choice of capping agent is typically determined by its impact on the stability of the AgNPs [37]. Utilizing green chemistry principles in the synthesis of AgNPs using plant extracts allows for the selection of phytochemical components that act as both reducing and covering agents. This approach enables control over the size and stabilization of the nanoparticles. In the bottom-up synthesis method of AgNPs, capping agents are often employed to regulate particle size, agglomeration, and morphology. These agents, characterized by specific functional groups, can bind to the surface of synthesized AgNPs, thereby reducing surface energy and preventing grain growth and particle agglomeration. Additionally, capping agents may impart electrostatic and steric stabilization effects, aiding in the dispersion of AgNPs in suspensions [38]. In our case the stability of silver nanoparticles (AgNPs) synthesized using methanolic and aqueous extracts at pH 2, 4, and 7, can be attributed to several factors:

1. Surface charge: At lower pH levels (pH 2 and 4), the surface charge of AgNPs tends to be positive due to protonation of surface functional groups. This positive charge helps repel nanoparticles from each other, preventing aggregation and maintaining stability. At pH 7, the surface charge may be slightly negative, but still sufficient to stabilize the nanoparticles through electrostatic repulsion. However, at higher pH levels (>7), deprotonation of functional groups on the nanoparticle surface leads to a decrease in positive charge, reducing electrostatic repulsion between particles and promoting aggregation.

2. Surface chemistry: The surface chemistry of AgNPs is influenced by the pH of the surrounding environment. At acidic pH, protonation of surface functional groups may

enhance the adsorption of stabilizing agents present in the extracts onto the nanoparticle surface, further stabilizing them against aggregation. However, at alkaline pH, deprotonation of these functional groups weakens the interaction between stabilizing agents and nanoparticles, leading to destabilization and aggregation.

3. Dissolution of silver ions: At high pH levels, there is an increased tendency for silver ions to undergo hydrolysis and form insoluble silver hydroxide or silver oxide species. This process can result in the dissolution of AgNPs and subsequent loss of stability.

Our findings are consistent with those of Ndikau et al. [39] and Khan et al. [40], which saw maxima at 433 nm and 424 nm, respectively. Due to Plasmon resonance excitation, the surface of silver nanoparticles often exhibits recognizable peaks between 400 nm and 500 nm [31].

**3.1.2. Analysis of functional groups using FTIR.** Using Fourier Transform Using infrared analysis, the functional groups involved in reducing and capping silver nanoparticles were identified. Based on the FT-IR measurements, both main and minor peaks were determined.

Possibly owing to O-H or N-H stretching vibrations of polyphenols, flavonoids, and alkaloids, the peaks at 3348.19 cm-1 moved to 3325.05 cm-1, and 3355.91 cm-1 shifted to 3348.19 cm-1 in the AgNPs (Table 1). Khanal et al. obtained a similar outcome [41]. Initial phytochemical data supported the findings because polyphenols are known to be natural reducing agents [42].

As a result of carboxylic acids, the spectral bands between 3300 and 2500 cm-1 have been attributed to O-H stretching. Similar to the findings published by Ahmad et al. [25], we observed a peak at 2939.31 cm-1, 2931,59 cm-1, and 2923.28 cm-1 (Table 1).

The band 1589.23 cm1 was detected between 1650 and 1585 cm-1 (Table 1), and this may be related to the N-H bending of 1˚ amine. Comparatively, the findings of these investigations were comparable to those published by Ahmad et al. [25], Hanna et al. [43], and other researchers, Pungle et al. [19]. C = C stretching vibrations attributed the peak at 1411.79 cm-1 to the aromatic compounds.

The detected band at 1110.92 cm-1 is due to the C-N stretch of an aliphatic amine group in the protein [44]. Simultaneously, many spectral bands, 1110.92, 1226.64, 1110.93, 1072.35, 1319.22, 1211.21, 1072.21 cm-1, resulting from C-H stretch owing to alcohols, carboxylic acids, esters, and ethers, as they lie between 1320 and 1000 cm-1, were detected.

Due to alkene and C–Br stretching due to alkyl halides, C-H bending corresponds to spectral bands at 987.05, 918.05, 848.02, 702.04, 594.03, and 601.75 cm -1, respectively. As shown by FTIR analysis, polyphenols, alkaloids, flavonoids, and proteins have a crucial role in decreasing, capping, and stabilizing biosynthesized AgNPs. Proteins contribute to the

**Table 1. FTIR analysis of methanol and aqueous extract and its mediated silver nanoparticles.**

| Frequency (cm$^1$) | Chemical bond | Assignment | Extracts | AgNPs |
|---|---|---|---|---|
| 3500–3200(s,b) | O–H stretch, N–H Stretch | Alcohols, phenols flavonoids/ alkaloids | 3348.19, 3355,91 | 3325.05, 3348.19 |
| 3300-2500(m) | O–H stretch | Carboxylic acids | 2923.28, 2939.31 | 2939.31, 2931,59 |
| 1650–1585(m) | N–H bend | 1 amines | 1589.23, 1589.23 | 1589.23, 1589.23 |
| 1500-1400(m) | C = C stretch | Aromatics | 1411,79 | |
| 1335–1250(s) | C–N stretch | Aromatic amines | 1326.94 | 1319.22, 1326.93 |
| 1320–1000(s) | C–H stretch | Alcohols, carboxylic acids, esters, ethers | 1110.92, 1072.35, 1226.64, 1110.93, 1072.35 | 1319.22, 1211.21, 1110.92, 1072.21, 1072.35 |
| 1000–650(s) | = C–H bend | Alkenes | 910.34, 694.33, 709.76 | 987.05, 918.05, 848.02, 702.04 |
| 640–515(m) | = C-Br | Alkyl halides | 609.86, 601.75 | 594.03, 601.75 |

reduction, stabilization, and prevention of nanoparticle aggregation [45,46]. This reinforces the stability, high surface charge, and superb colloidal nature of dynamic light scattering.

**3.1.3. Dynamic light scattering.** Fig 8A and 8B depict the results for the average particle size and stability of produced AgNPs from the methanolic extract, while Fig 9A and 9B show the average particle size and stability of produced AgNPs from the aqueous extract, respectively.

Nanoparticle sizes are typically between 1 and 100 nm [43]. The nanoparticles' positivity and negativity determine their aggregative quality. When two charges coexist, they tend to

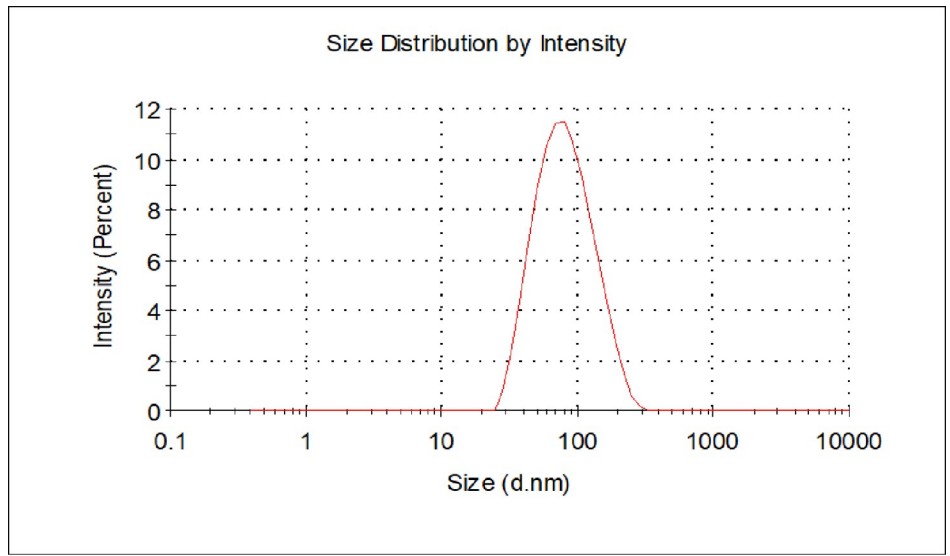

a.

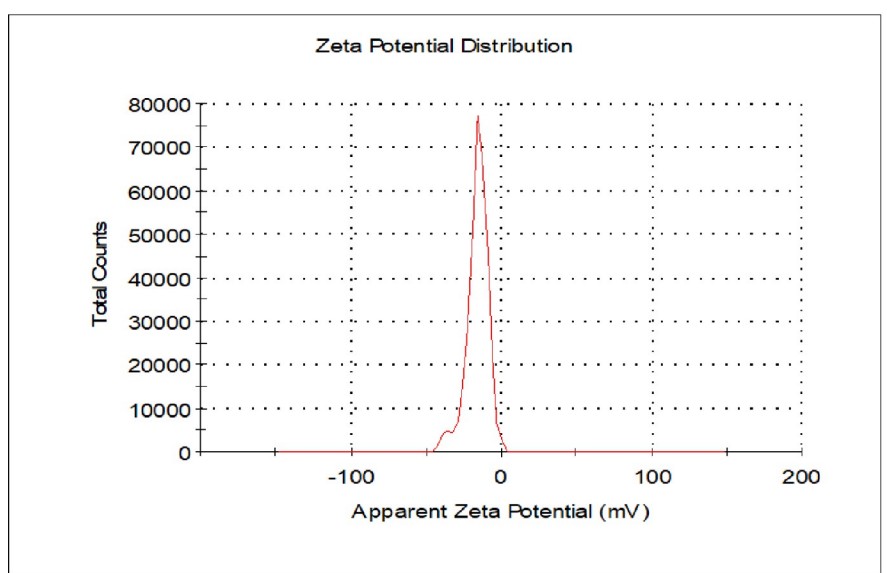

b.

**Fig 8.** a. size distribution of AgNPs of *Ocimum kilimandscharicum* methanolic leaf extracts. b. zeta potential distribution of AgNPs of *Ocimum kilimandscharicum* methanolic leaf extracts.

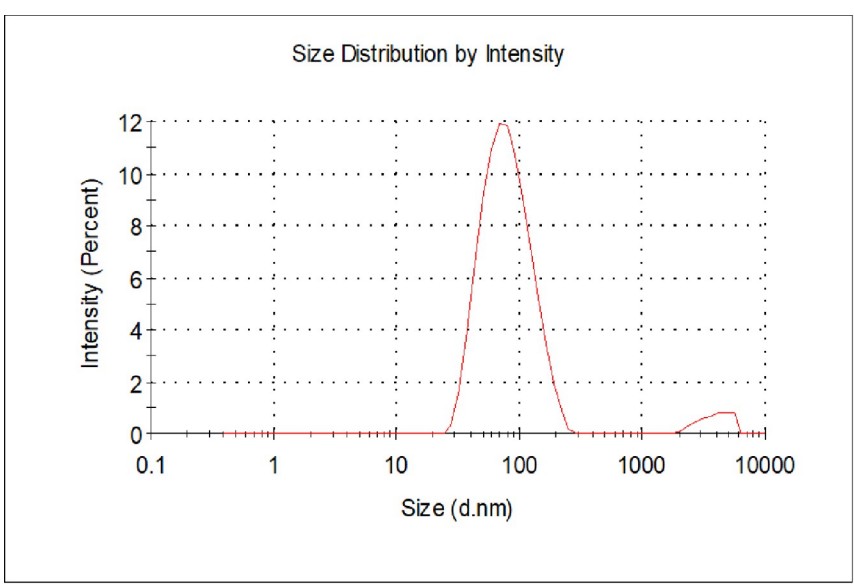

**a.**

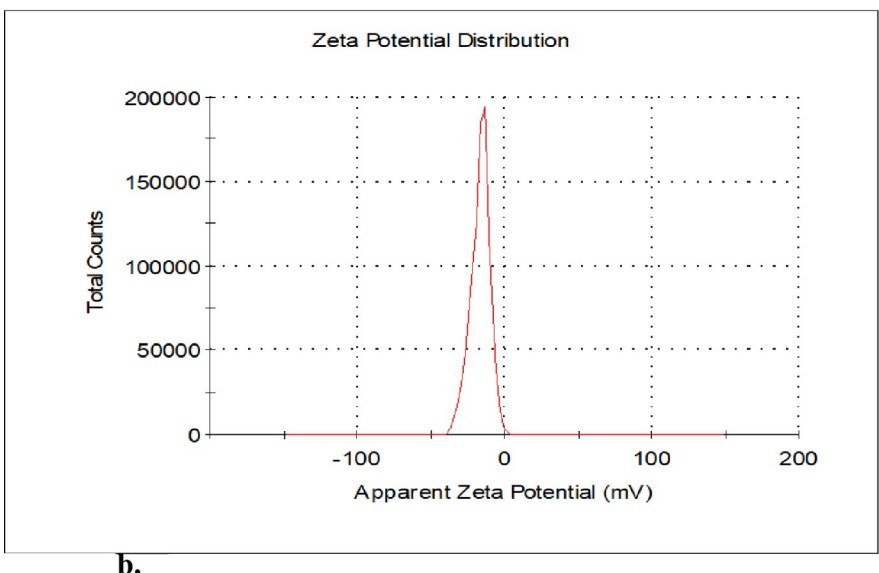

**b.**

**Fig 9.** a. size distribution of AgNPs of *Ocimum kilimandscharicum* aqueous leaf extract. b. zeta potential distribution of AgNPs of *Ocimum kilimandscharicum* aqueous leaf extracts.

attract one another, resulting in their accumulation [47]. The *O. kilimandscharicum*-mediated AgNPs from the methanolic extract had an average particle size of 69.72 ± 0.45 nm, a polydispersity index of 0.27 ± 0.026, and a zeta potential of -17.6 ± 1.89 mV. For the *O. kilimandscharicum*-mediated AgNPs from the aqueous extract, the average particle size was 70.14 ± 1.15 nm, the polydispersity index was 0.27 ± 0.005, and the zeta potential was -18.1 ± 2.47 mV as determined by dynamic light scattering (Table 2, respectively). Negative charges suggested that the biosynthesized nanoparticles lacked clustering and clumping characteristics [48]. Studies have

**Table 2. Characterization of the silver nano formulated from methanolic and aqueous extracts of *Ocimum kilimandscharicum* leaves by zeta.**

| Solvent extract | Size distribution (d. nm) ± SE | Polydispersity index (PDI) ± SE | Zeta potential distribution (mV) ± SE |
|---|---|---|---|
| Methanol | 69.72 ± 0.45 | 0.27 ± 0.026 | -17.6 ± 1.95 |
| Water | 70.14 ± 1.15 | 0.27 ± 0.005 | -18.1 ± 2.47 |

demonstrated how a positive charge of + 5.68 mV resulted in agglomerative solid potential [49]. Similar results were also reported; -24.6 mV [50], -30 mV [51], -22.3 mV [52], and -22.7 mV [48]. The polydispersity index (PDI) for monodispersed particles ranges from 0.01 to 0.70, while polydispersed particles have more than seven (7) polydispersity index values [48]. Our findings indicated nanoparticles to be promising in terms of particle size, stability, and dispersion.

## 3.2. Antimicrobial screening of the extracts and their silver nanoparticles

**3.2.1. Zone of inhibition (ZOI).** In the present study, we evaluated the antibacterial activity of silver nanoparticles synthesized from the methanolic and aqueous leaf extracts of *O. kilimandscharicum* and its crude extract against Gram-negative (*Escherichia coli ATCC 25922, Salmonella choleraesuius ATCC 10708*) and Gram-positive bacteria (*Staphylococcus aureus ATCC 25923*). Figs 10 and 11, and Table 3 show the findings, respectively.

Values are the mean ± SE. Values with the same superscript letters designate no significant differences.

The findings of the zone of inhibition were represented as mm. The data were presented as Mean ± SE of triplicate, as shown in Table 3. Silver nanoparticles made from aqueous extract had the highest zone of inhibition (16 ± 1 mm), whereas water and methanolic extracts had the lowest (8 ± 1 mm). Silver nanoparticles are more effective than crude extracts, as shown in Table 3, where the lowest zone of inhibition of silver nanoparticles is 12 ± 1 mm, and the one for crude extracts is 8 ± 1 mm. Moreover, silver nanoparticles are more effective against Gram-negative bacteria, *Escherichia coli*, and *Salmonella choleraesuius* (zone of inhibition between 15 to 16 ± 1 mm). In contrast, they are the least effective on Gram-positive bacteria,

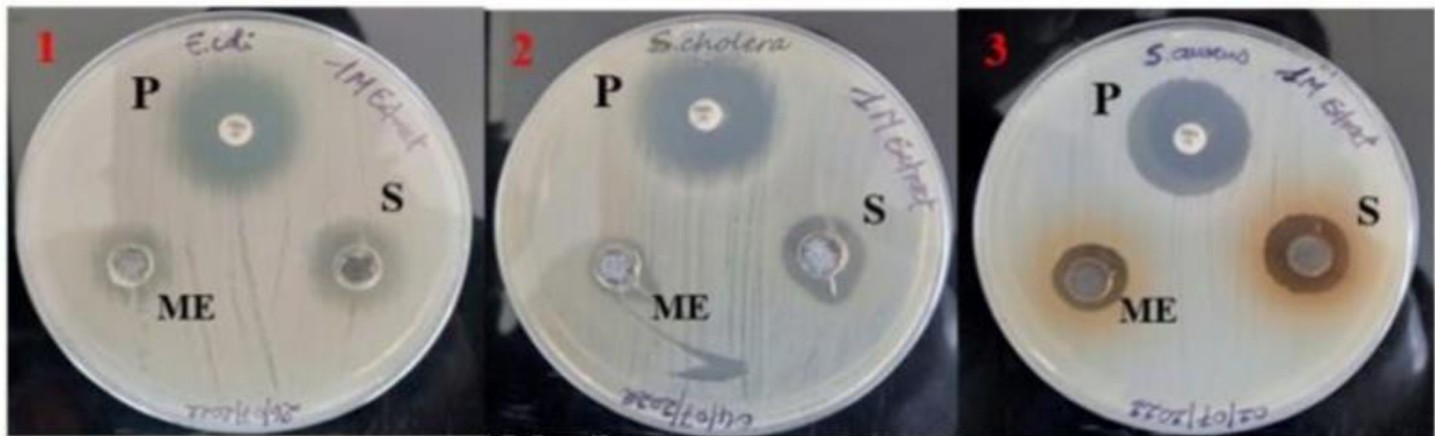

**Fig 10. Zones of inhibition for the methanolic extract and silver nanoparticles of the methanolic extract of *O. kilimandscharicum*.** 1 = Escherichia coli; 2 = Salmonella choleraesuius; 3 = Staphylococcus aureus; S = silver nanoparticles; ME = methanolic extract; P = positive control (gentamicin); 1M = methanolic extract of *O. kilimandscharicum*. Each experiment was done in triplicate.

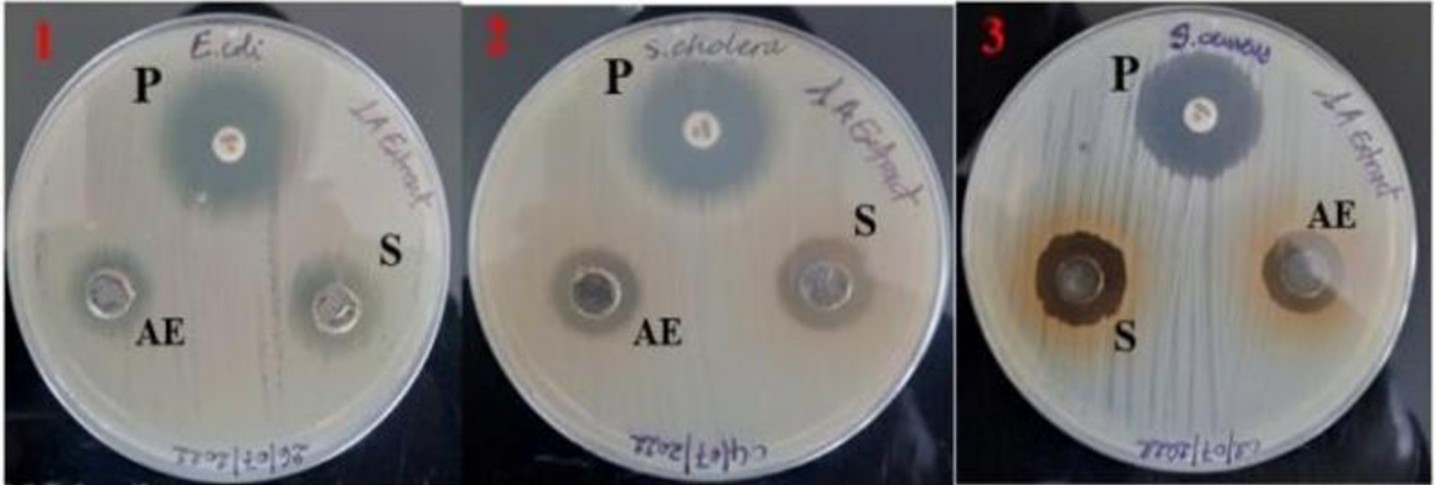

**Fig 11. Zones of inhibition for the aqueous extract and silver nanoparticles of the aqueous extract of *O. kilimandscharicum*.** 1 = Escherichia coli; 2 = Salmonella choleraesuius; 3 = Staphylococcus aureus; S = silver nanoparticles; AE = aqueous extract; P = positive control (gentamicin); 1A = aqueous extract of. *O. kilimandscharicum*. Each experiment was done in triplicate.

*Staphylococcus aureus* (zone of inhibition between 12 at 13 ± 1 mm. ANOVA of aqueous silver nanoparticles (F (2,6) = 13.00, p<0.05) in Table 3 revealed that there was a statistically significant difference between the groups. The superscripted letters in Table 3 indicate significant differences between means.

Several publications describe the processes by which silver nanoparticles function, such as alteration of membrane permeability [28,41], bacterial protein precipitation [53,54], tiny size and vast surface area to attach to the cell wall [44], DNA cleavage [53], production of free radicals, and electrostatic attraction [47,55].

Mariselvam *et al.* [56] and Numan *et al.* [57] consider zones of inhibition with sizes of < 9 mm, 9–12 mm, and 13–18 mm inactive, moderately active, and active, respectively. This study revealed that silver nanoparticles produced from methanolic and aqueous extracts of *O. kilimandscharicum* and crude extracts have significant antibacterial activity against Gram-positive and negative bacteria (Fig 12 and Table 3). Different zones of inhibition demonstrated that

**Table 3. Zone of inhibition against some bacteria strains by methanol extract, aqueous extract and AgNPs of methanol extract, AgNPs of aqueous extract of *Ocimum kilimandscharicum*.**

| Test organisms | Diameters of zones of inhibition in mm ± SE; concentration of extracts tested (50mg/ml) | | | | |
|---|---|---|---|---|---|
| | Methanol AgNPs | Methanol crude | Aqueous AgNPs | Aqueous crude | Gentamycin (10mg/ml) |
| *Escherichia coli* | 15 ± 1[c] | 10 ± 1[e] | 15 ± 1[c] | 9 ± 1e, f | 23 ± 0[b] |
| *Salmonella choleraesuius* | 15 ± 1[c] | 9 ± 1e, f | 16 ± 1[c] | 10 ± 1[e] | 25 ± 1[a] |
| *Staphylococcus aureus* | 13 ± 1[d] | 8 ± 1[f] | 12 ± 1[d] | 8 ± 1[f] | 24 ± 1a, b |
| Mean | 14.33 ± 1.15 | 9 ± 1 | 14.33 ± 2.08 | 9 ± 1 | 24 ± 1 |
| F-Value | 4.00 | 3.00 | 13.00 | 3.00 | 4.50 |
| P-Value | >0.05 | >0.05 | <0.05 | >0.05 | >0.05 |
| LSD (0.05) | 1.99 | 1.99 | 1.99 | 1.99 | 1.63 |
| CV, % | 2.44 | 2.44 | 2.44 | 2.44 | 2.44 |

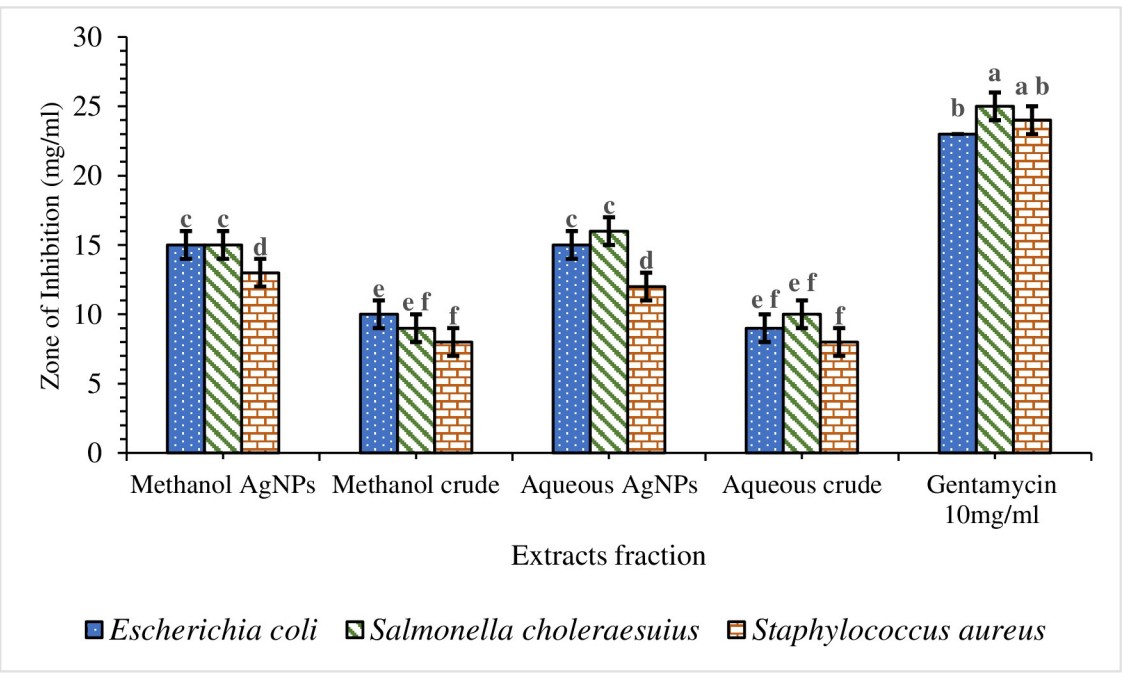

**Fig 12. The zone of inhibition for the aqueous and methanolic extracts and their silver nanoparticles of *O. kilimandscharicum* against the indicated bacteria.** Values with the same letters designate no significant differences.

silver nanoparticles were more efficient against Gram-negative bacteria (*S. choleraesuius* and *E. coli*) than against Gram-positive bacteria (*S. aureus*) (Fig 12).

Baran *et al*. [47], Khan *et al*. [40], Ahmed *et al*. [58], and Khanal *et al*. [41] all reported identical findings. This may be due to changes in cell wall composition or stiffness between Gram-positive and Gram-negative bacteria [59,60].

In contrast, Phuyal *et al*. [44], and Kaur *et al*. [61] demonstrated more antibacterial activity against Gram-positive bacteria (*S. aureus*) than against Gram-negative bacteria (*E. coli* and *S. choleraesuius*).

Khalil *et al*. [62] presented intriguing data about the antibacterial activity of the crude extract of *Ziziphus nummularia* fruit compared to its mediated silver nanoparticles, which aligns with our results. The diameters of the inhibitory zones for the crude aqueous extract of *Ziziphus nummularia* fruit were 11.66 mm for *S. aureus* and 6.30 mm for *E. coli*. For silver nanoparticles, however, it climbed to 16.66 mm for *S. aureus* and 18.66 mm for *E. coli*, respectively [62].

Khanal *et al*. [41] reported root extracts of *Rubus ellipticus* with zone of inhibition values of 10 mm (*E. coli*), 9 mm (*S. aureus*), 10 mm (*K. pneumoniae*), and 11 mm (*E. faecalis*). In contrast, the root extracts of *Rubus ellipticus*-mediated silver nanoparticles were 13 mm (*E. coli*), and 12 (*E. faecalis*). This proved that the antibacterial properties of silver nanoparticles were greater than those of the crude extract [41].

**3.2.2. Minimum inhibitory concentration (MIC) and minimum bactericidal concentration (MBC).** MIC is the lowest concentration at which no colour change occurs [63,64], whereas MBC is the lowest concentration at which no growth is seen on a petri dish [65].

The findings of the MIC and MBC were represented as mg/ml. The data were presented as Mean ± SE of triplicate, as shown in Table 4. Figs 13, 14, 15A, 15B, 16A, 16B, 17A, 17B, 18 and

**Table 4. MIC (mg/ml) and MBC (mg/ml) ± SE against some bacteria strains by methanol extract, aqueous extract and AgNPs of methanol extract, AgNPs of aqueous extract of Ocimum kilimandscharicum.**

| Methanol AgNPs | | | Methanol crude | | Aqueous AgNPs | | Aqueous crude | |
|---|---|---|---|---|---|---|---|---|
| **Test organisms** | | | | | | | | |
| | **MIC** | **MBC** | **MIC** | **MBC** | **MIC** | **MBC** | **MIC** | **MBC** |
| *Escherichia coli* | 6.25 ± 0.0[d] | 12.5 ± 0.0[c] | 25 ± 0.0[b] | 50 ± 0.0[a] | 6.25 ± 0.0[d] | 12.5 ± 0.0[c] | 25 ± 0.0[b] | 50 ± 0.0[a] |
| *Salmonella choleraesuius* | 6.25 ± 0.0[d] | 12.5 ± 0.0[c] | 25 ± 0.0[b] | 50 ± 0.0[a] | 6.25 ± 0.0[d] | 12.5 ± 0.0[c] | 25 ± 0.0[b] | 50 ± 0.0[a] |
| *Staphylococcus aureus* | 6.25 ± 0.0[d] | 25 ± 0.0[b] | 25 ± 0.0[b] | 50 ± 0.0[a] | 6.25 ± 0.0[d] | 25 ± 0.0[b] | 25 ± 0.0[b] | 50 ± 0.0[a] |
| Mean | 6.25 ± 0.00 | 16.66 ± 7.21 | 25 ± 0.0 | 50 ± 0.0 | 6.25 ± 0.0 | 16.66 ± 7.21 | 25 ± 0.0 | 50 ± 0.0 |
| F-Value | - | Infinity | - | - | - | Infinity | - | - |
| P-Value | - | <0.0001 | - | - | - | <0.0001 | - | - |
| LSD (0.05) | 0 | 0 | 0 | 0 | 0 | 0 | 0 | 0 |
| CV, % | 2.44 | 2.44 | 2.44 | 2.44 | 2.44 | 2.44 | 2.44 | 2.44 |

Values are the mean ± SE. Values with the same superscript letters designate no significant differences.

Table 4 show the findings, respectively. Silver nanoparticles made from aqueous and methanolic extracts had the lowest MIC (6.25 ± 0.00 mg/ml), whereas water and methanolic extracts had the highest MIC (25 ± 0.00 mg/ml). Silver nanoparticles made from aqueous and methanolic extracts had the lowest MBC (12.5 ± 0.00 and 25 ± 0.00 mg/ml), whereas water and methanolic extracts had the highest MBC (50 ± 0.00 mg/ml). Silver nanoparticles are more effective than crude extracts, as shown in Table 4, where the MIC of silver nanoparticles is 6.25 ± 0.00 mg/ml, and the one for crude extract is 25 ± 0.00 mg/ml. As we said above, silver nanoparticles are more effective on Gram- negative bacteria, *Escherichia coli* and *Salmonella*

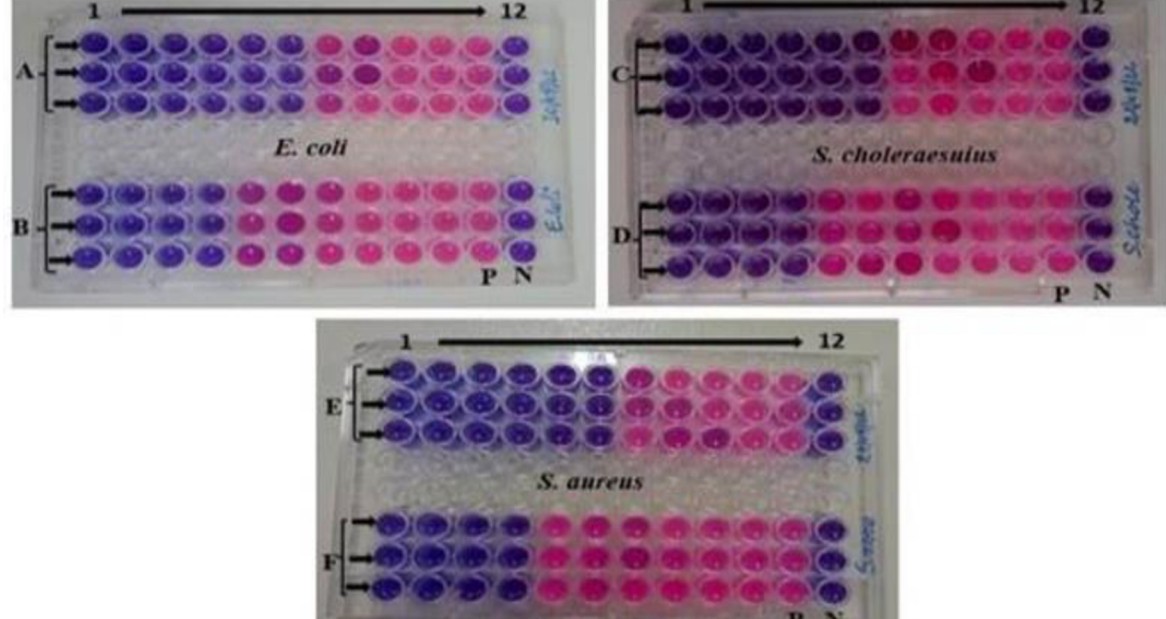

**Fig 13. Minimum inhibitory concentrations of the methanolic extract and silver nanoparticles of *O. kilimandscharicum* against the indicated bacteria.** A, C & E = silver nanoparticles; B, D & F = methanolic extract; 1–12 = wells; P = positive control (bacteria without extract); N = negative control (Mueller Hinton Broth without bacteria).

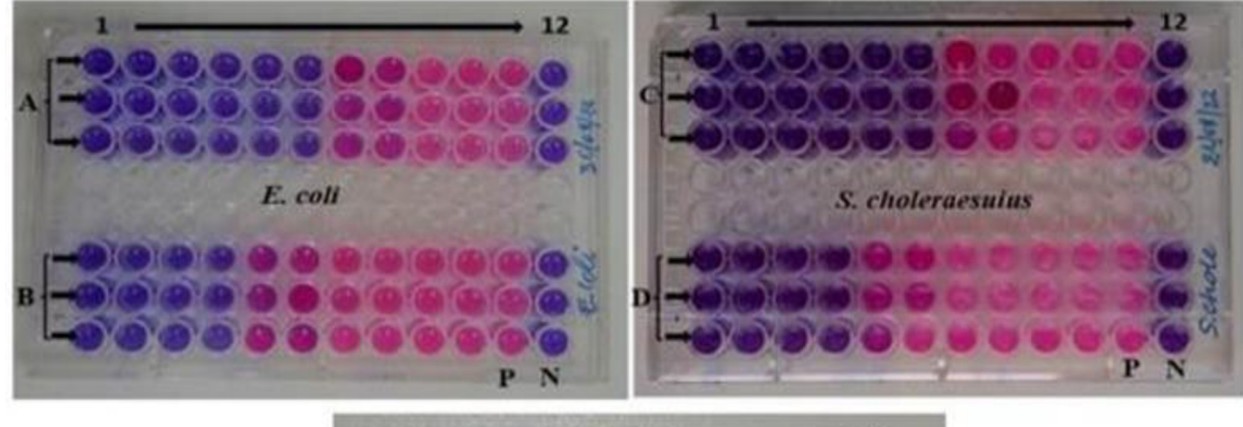

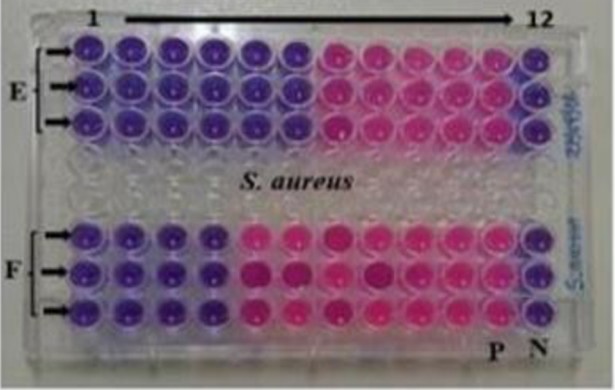

**Fig 14. Minimum inhibitory concentrations of the aqueous extract and silver nanoparticles of O. kilimandscharicum against the indicated bacteria.** A, C & E = silver nanoparticles; B, D & F = aqueous extract; 1–12 = wells; P = positive control (bacteria without extract); N = negative control (Mueller Hinton Broth without bacteria).

*choleraesuius* (MBC 12.5 ± 0.00 mg/ml). In contrast, they are the least effective on Gram-positive bacteria, *Staphylococcus aureus* (MBC, 25 ± 0.00 mg/ml). Only ANOVA of MBCs' aqueous and methanol silver nanoparticles (F (2,6) = Infinity, p<0.0001) in Table 4 revealed that there was a statistically significant difference between the groups. The superscripted letters in Table 4 indicate significant differences between means.

Abdellatif *et al*. [66] showed a similar scenario employing an aqueous extract of *Thymus vulgaris* leaves, *Zingiber officinale* roots, and their silver nanoparticles against the *S. aureus* clinical strain. The MBC value of aqueous *Thymus vulgaris* leaf extract was 0.2825 mg/ml, which decreased to 0.0706 mg/ml in silver nanoparticles. For aqueous preparations of *Zingiber officinale* root extract, the concentration decreased from 2,2600 mg/ml to 0.5650 mg/ml [66].

Several research investigations, such as Periasamy *et al*. [67], Periyasami *et al*. [68], Phuyal *et al*. [44], and Khanal *et al*. [41], have shown the occurrence of silver nanoparticles with strong biological activity. The process of phytochemical screening involves both qualitative and quantitative analysis of chemical classes present in ethnomedicinal plant species. Utilizing methanolic and aqueous extraction methods in conjunction with gas chromatography-mass spectrometry (GC-MS) has emerged as a prevalent technique for identifying phytochemicals with clinical significance [69,70]. Employing this methodology, we ascertain the primary bioactive compounds within O. kilimandscharicum leaves. Our analysis revealed the existence of alkaloids, phenolics, flavonoids, and saponins in both aqueous and methanolic extracts of *O.*

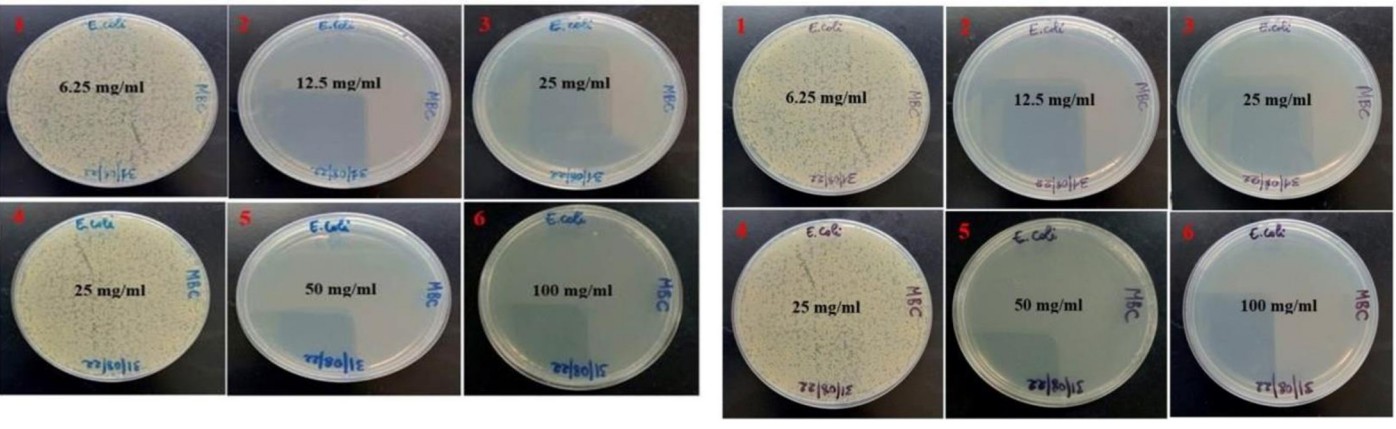

**Fig 15.** a. Minimum bactericidal concentrations of the methanolic extract and silver nanoparticles of *O. kilimandscharicum* against *E. coli*. 1, 2 & 3 = silver nanoparticles; 4, 5 & 6 = methanolic extract, MBC = Minimum Bactericidal Concentration. b. Minimum bactericidal concentrations of the aqueous extract and silver nanoparticles of *O. kilimandscharicum* against *E. coli*. 1, 2 & 3 = silver nanoparticles; 4, 5 & 6 = aqueous extract, MBC = Minimum Bactericidal Concentration.

*kilimandscharicum* leaves. On the other hand, our findings suggest the absence of tannins in aqueous leaf extract, which is not surprising given that the content of tannins are optimal in hydro-alcoholic extracts [1]. Our research uncovered significantly elevated levels of phenolics and tannins in the methanol extract compared to the aqueous extract. Hence, we propose the utilization of the methanolic extract for forthcoming nano formulation investigations [1].

## 4. Conclusion

This work describes the manufacture of silver nanoparticles from aqueous and methanolic O. kilimandscharicum leaf extract. UV–Vis revealed silver nanoparticles' surface plasmon

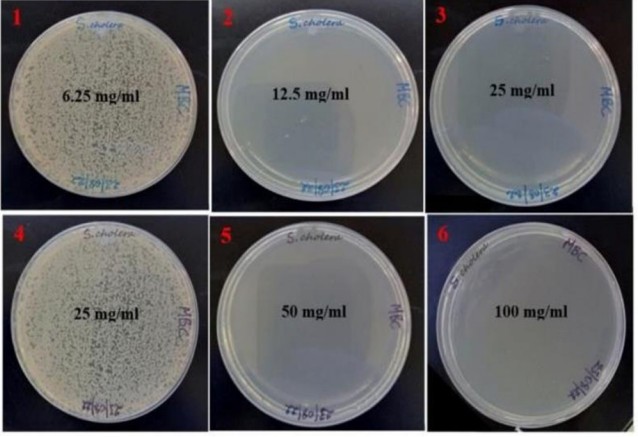

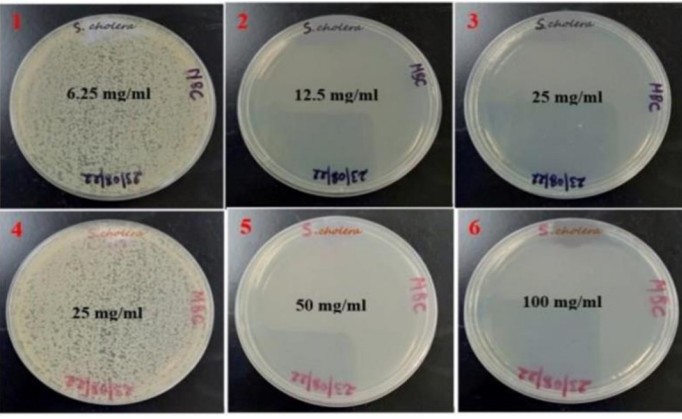

**Fig 16.** a. Minimum bactericidal concentrations of the methanolic extract and silver nanoparticles of *O. kilimandscharicum* against *Salmonella choleraesuius*. 1, 2 & 3 = silver nanoparticles; 4, 5 & 6 = methanolic extract, MBC = Minimum Bactericidal Concentration. b. Minimum bactericidal concentrations of the aqueous extract and silver nanoparticles of *O. kilimandscharicum* against *Salmonella choleraesuius*. 1, 2 & 3 = silver nanoparticles; 4, 5 & 6 = aqueous extract, MBC = Minimum Bactericidal Concentration.

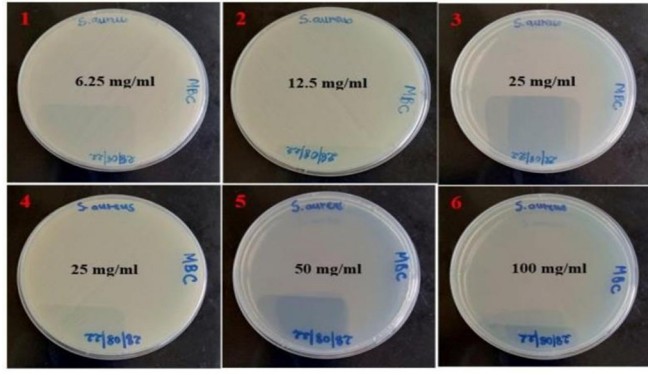
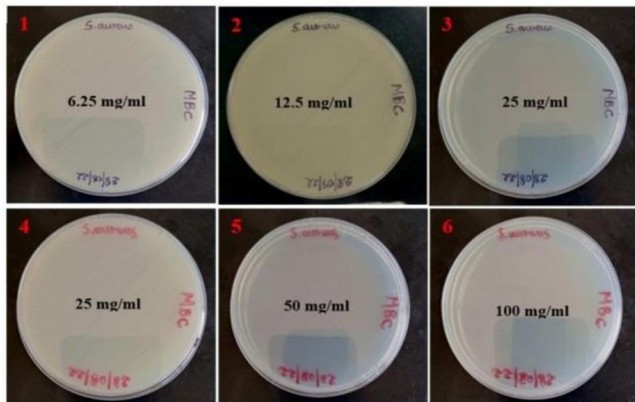

**Fig 17.** a. Minimum bactericidal concentrations of the methanolic extract and silver nanoparticles of *O. kilimandscharicum* against *Staphylococcus aureus*. 1, 2 & 3 = silver nanoparticles; 4, 5 & 6 = methanolic extract, MBC = Minimum Bactericidal Concentration. b. Minimum bactericidal concentrations of the aqueous extract and silver nanoparticles of *O. kilimandscharicum* against *Staphylococcus aureus*. 1, 2 & 3 = silver nanoparticles; 4, 5 & 6 = aqueous extract, MBC = Minimum Bactericidal Concentration.

resonance. Biosynthesized AgNPs demonstrated significant antibacterial action against all pathogens. To combat rising concerns, AgNPs will make a considerable contribution to anti-microbial agent research. We suggest that the work can be continued to examine the *in vitro* and *in vivo* toxicity of silver nanoparticles generated sustainably.

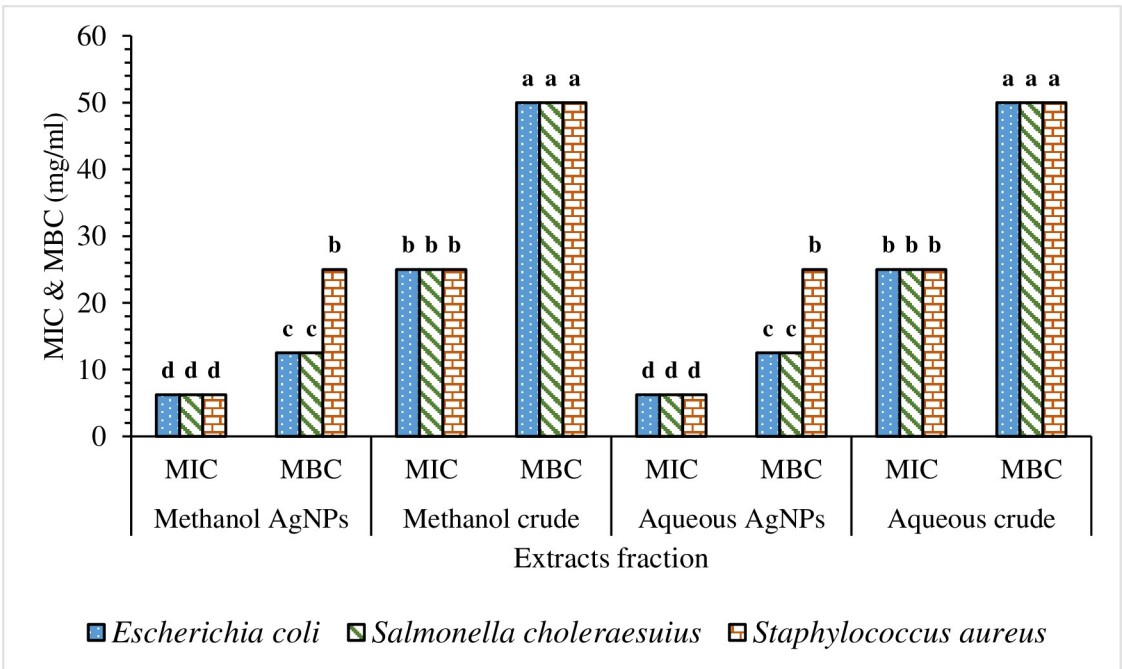

**Fig 18. The MIC and MBC for the aqueous and methanolic extract and their silver nanoparticles of *O. kilimandscharicum* against the indicated bacteria.** Values with the same letters designate no significant differences.

## Supporting information

**S1 File.**
(ZIP)

## Acknowledgments

The research from which this dataset was obtained was funded by the Pan African University Institute for Basic Sciences, Technology and Innovation Doctoral grant to HSO, MB401-0005/19. The Pan African University of Basic Science, Technology, and Innovation provided the facilities and laboratory support necessary for the execution of this experiment, which the authors gratefully recognize. They are quite appreciative of the African Union's assistance with research money. The bacteria strains were generously provided by the Kenya Medical Research Institute (KEMRI), for which the authors are very grateful.

## Author Contributions

**Conceptualization:** Horyomba Siaka Ouandaogo, Eddy Odari.

**Data curation:** Horyomba Siaka Ouandaogo.

**Formal analysis:** Horyomba Siaka Ouandaogo, Souleymane Diallo.

**Investigation:** Horyomba Siaka Ouandaogo.

**Methodology:** Horyomba Siaka Ouandaogo.

**Project administration:** Horyomba Siaka Ouandaogo.

**Resources:** Horyomba Siaka Ouandaogo.

**Software:** Horyomba Siaka Ouandaogo.

**Supervision:** Souleymane Diallo, Eddy Odari, Johnson Kinyu.

**Visualization:** Horyomba Siaka Ouandaogo.

**Writing – original draft:** Horyomba Siaka Ouandaogo.

**Writing – review & editing:** Horyomba Siaka Ouandaogo.

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
