## [Decision Letter · Decision Letter 0]

18 Dec 2023

PONE-D-23-37831Silver nanoparticle biosynthesis utilizing Ocimum kilimandscharicum leaf extract and assessment of its antibacterial activity against certain chosen bacteriaPLOS ONE

Dear Dr. Ouandaogo,

Thank you for submitting your manuscript to PLOS ONE. After careful consideration, we feel that it has merit but does not fully meet PLOS ONE’s publication criteria as it currently stands. Therefore, we invite you to submit a revised version of the manuscript that addresses the points raised during the review process.

We look forward to receiving your revised manuscript.

Kind regards,

Jorddy Neves Cruz

Academic Editor

PLOS ONE

Journal Requirements:

Reviewers' comments:

Reviewer's Responses to Questions

**Comments to the Author**

1. Is the manuscript technically sound, and do the data support the conclusions?

Reviewer #1: Yes

Reviewer #2: Yes

Reviewer #3: Partly

Reviewer #4: Yes

2. Has the statistical analysis been performed appropriately and rigorously? 

Reviewer #1: Yes

Reviewer #2: Yes

Reviewer #3: N/A

Reviewer #4: Yes

3. Have the authors made all data underlying the findings in their manuscript fully available?

Reviewer #1: Yes

Reviewer #2: Yes

Reviewer #3: No

Reviewer #4: Yes

4. Is the manuscript presented in an intelligible fashion and written in standard English?

Reviewer #1: Yes

Reviewer #2: No

Reviewer #3: No

Reviewer #4: Yes

5. Review Comments to the Author

Reviewer #1: Reviewer’s Constructive Feedback to the Authors:

I have had the opportunity to review your study titled "Silver nanoparticle biosynthesis utilizing Ocimum kilimandscharicum leaf extract and assessment of its antibacterial activity against certain chosen bacteria." The abstract provides a clear overview of the study, which focuses on the use of Ocimum kilimandscharicum for the biological production of silver nanoparticles (AgNPs) and their potential antibacterial properties. There is no specific fault in the given text. However, here are some comments and suggestions:

1. There is inconsistency in the use of tenses. For example, the phrase "is becoming increasingly important" is in the present continuous tense, while the rest of the abstract is predominantly in the past tense.

2. A few typos in the document, such as aluminium instead of aluminum, space problem, comma and fullstop issues. Additionally, there are some other grammatical errors that may need attention.

3. The results and discussion sections are written together in this manuscript. I don't know if it falls within the rules of the journal or not. It is preferable to place this in different parts.

4. Picture quality should be increased, especially for Fig. 19a, b; For MIC and MBC they can include the picture of broth culture.

In this study the authors focuses on the synthesis of silver nanoparticles (AgNPs) using aqueous and methanolic extracts of Ocimum kilimandscharicum leaves. The UV-Vis analysis confirmed the presence of silver nanoparticles through the observation of their surface plasmon resonance. The biosynthesized AgNPs exhibited significant antibacterial activity against various pathogens. Considering the growing concerns regarding microbial resistance, AgNPs hold great potential as antimicrobial agents. Further research is recommended to investigate the in vitro and in vivo toxicity of these sustainable silver nanoparticles

Reviewer #2: The author has written a manuscript titled Silver nanoparticle biosynthesis utilizing Ocimum kilimandscharicum leaf extract and assessment of its antibacterial activity against certain chosen bacteria.

1) manusript came out well

2) In introduction, add some more points and literature review

3) In fugure 2 Menthanol crude extrat graph is above the range, do it once again and add in the graph

4) You have added plain silver meance nano or silver nitrate; please express correctly

5) Figure 3 only writes an OD in nm not an absorbance.

6) You have performed the experiment with aquous extract and methanol extract. need comparative study on all phytochemical analyses and which one is best to isolate bioactive molecules

7) In figure 6, why is there is peak in 360nm and what is the significance?

8) please name the peaks

9) Figure 8 and 9 what is a comparison interpretation?add

10) only one application is done; somemore more applications like anti-cancer activity, environmental study , antioxidant activity etc

Reviewer #3: Dear editor and author

Overall this is a good article that presents a novel synthesis strategy and good characterization techniques. However, I would like to make some suggestions.

I am attaching the manuscript with some comments I would like you to address.

1.There are lots of typo and grammatical mistakes in the manuscript making the manuscript less interesting to read.

2.IN FTIR studies, these functional groups indicated flavonoids, phenols, and ascorbic acid in Ocimum kilimandscharicum leaf extract-----FTIR can indicate only functional groups and not presence of compounds or metabolites.

3.XRD and EDX, and TEM added the manuscript.

4.The novelty of the research work in terms of prior art available has to be brought out.

Reviewer #4: The authors did interesting research work with a lot of results. thus, suggestion to combine or choose either figures or tables for some results. for example, FT-IR spectra are not needed when the table resumes all the results for it.

6. PLOS authors have the option to publish the peer review history of their article (what does this mean?). If published, this will include your full peer review and any attached files.

Reviewer #1: **Yes: **Shahina Akter, Principal Scientific Officer, Bangladesh Council of Scientific and Industrial Research (BCSIR), Email- shupty2010@gmail.com

Reviewer #2: No

Reviewer #3: No

Reviewer #4: **Yes: **Amadou Issoufou

---

## [Author Response · Author response to Decision Letter 0]

16 Apr 2024

Reviewer(s)' Comments to Author:

Reviewer: 1

Comments:

I have had the opportunity to review your study titled "Silver nanoparticle biosynthesis utilizing Ocimum kilimandscharicum leaf extract and assessment of its antibacterial activity against certain chosen bacteria." The abstract provides a clear overview of the study, which focuses on the use of Ocimum kilimandscharicum for the biological production of silver nanoparticles (AgNPs) and their potential antibacterial properties. There is no specific fault in the given text.

Answer: Thank you for the appreciation

Questions:

There is inconsistency in the use of tenses. For example, the phrase "is becoming increasingly important" is in the present continuous tense, while the rest of the abstract is predominantly in the past tense.

Answer: We thank the reviewer for the insightful comments and suggestions. We improved the the use of tenses in the manuscript. Thank you.

A few typos in the document, such as aluminium instead of aluminum, space problem, comma and fullstop issues. Additionally, there are some other grammatical errors that may need attention.

Answer: We thank the reviewer for the insightful comments and suggestions. We improved the quality of the document using your comment. We corrected the word aluminum, resolved the space problem and the grammatical errors. Thank you.

The results and discussion sections are written together in this manuscript. I don't know if it falls within the rules of the journal or not. It is preferable to place this in different parts.

Answer: We thank the reviewer for the insightful comments and suggestions. We checked PlosOne’s many articles, we have seen that both are accepted. Thank you.

Picture quality should be increased, especially for Fig. 19a, b; For MIC and MBC they can include the picture of broth culture.

We thank the reviewer for the insightful comments and suggestions. We increased the quality of pictures. Thank you very much.

In this study the authors focuses on the synthesis of silver nanoparticles (AgNPs) using aqueous and methanolic extracts of Ocimum kilimandscharicum leaves. The UV-Vis analysis confirmed the presence of silver nanoparticles through the observation of their surface plasmon resonance. The biosynthesized AgNPs exhibited significant antibacterial activity against various pathogens. Considering the growing concerns regarding microbial resistance, AgNPs hold great potential as antimicrobial agents. Further research is recommended to investigate the in vitro and in vivo toxicity of these sustainable silver nanoparticles

Answer: We thank the reviewer for the insightful comments and suggestions. Definitely. Considering the growing concerns regarding microbial resistance, AgNPs hold great potential as antimicrobial agents. We can reassure you the next step of our research is what you suggested. In vitro and in vivo toxicity of this sustainable silver nanoparticles. Thank you very much.

Reviewer: 2

Questions:

Manuscript came out well.

Answer: Thank you for the appreciation.

In introduction, add some more points and literature review.

Answer: We thank the reviewer for the insightful comments and suggestions. We added some points and literature review.

In figure 2 Methanol crude extract graph is above the range, do it once again and add in the graph

Answer: We thank the reviewer for the insightful comments and suggestions. We did it many times but we still got the same result. I think this due to the high concentration of the phytochemical’s compounds in the Methanol crude extract.

You have added plain silver mean nano or silver nitrate; please express correctly.

Answer: We thank the reviewer for the insightful comments and suggestions. It is the silver nitrate. But when we say the pain silver solution it means the silver nitrate. Thank you very much.

Figure 3 only writes an OD in nm not an absorbance.

Answer: We thank the reviewer for the insightful comments and suggestions. In literature UV/Vis Spectra is express like that. Absorbance (a.u.) and wavelength (nm). Thank you very much. 

You have performed the experiment with aqueous extract and methanol extract. need comparative study on all phytochemical analyses and which one is best to isolate bioactive molecules.

Answer: We thank the reviewer for the insightful comments and suggestions. We did it in our previous study. We added it to the current manuscript. Thank you very much.

In figure 6, why is there is peak in 360nm and what is the significance?

Answer: We thank the reviewer for the insightful comments and suggestions. Certainly, the significance of the peak at 360 nm is undeniable. However, our research is centered on nanoparticles, which typically form within the wavelength range of 400 to 500 nm.

Figure 8 and 9 what is a comparison interpretation? Add

Answer: We thank the reviewer for the insightful comments and suggestions. One of the Reviewer suggested to delete these figures. Thank you very much. 

Only one application is done; some more applications like anti-cancer activity, environmental study, antioxidant activity.

Answer: We thank the reviewer for the insightful comments and suggestions. Definitely. Considering the growing concerns regarding microbial resistance, AgNPs hold great potential as antimicrobial agents. We can reassure you the next step of our research is what you suggested. To add more application like anti-cancer activity, environmental study, antioxidant activity, in vitro and in vivo toxicity. Thank you very much.

Reviewer: 3

Comments:

There are lots of typo and grammatical mistakes in the manuscript making the manuscript less interesting to read.

Answer: We thank the reviewer for the insightful comments and suggestions. We improved the quality of the document using your comment. We corrected the grammatical errors. Thank you.

In FTIR studies, these functional groups indicated flavonoids, phenols, and ascorbic acid in Ocimum kilimandscharicum leaf extract-----FTIR can indicate only functional groups and not presence of compounds or metabolites.

Answer: We thank the reviewer for the insightful comments and suggestions. 

XRD and EDX, and TEM added the manuscript.

Answer: We thank the reviewer for the insightful comments and suggestions. But we subbed them with the Dynamic Light Scattering using Malvern ZETASIZER NANO to determine the particle size, polydispersity index (PDI), and stability of biosynthesized AgNPs. And didn’t do it due to our limited budget. 

The novelty of the research work in terms of prior art available has to be brought out.

Answer: In the literature, there are several studies that aimed to evaluate the medicinal and insecticidal properties of plants from Ocimum genus. However, the Kenyan cultivars have not yet been fully studied. We expect that the preliminary phytochemical screening of Ocimum kilimandscharicum and silver nanoparticle biosynthesis utilizing Ocimum kilimandscharicum leaf extract and assessment of its antibacterial activity could pave the way to the efficient use of this plant in human and animal health.

Reviewer: 4

Comment:

The authors did interesting research work with a lot of results. thus, suggestion to combine or choose either figures or tables for some results. for example, FT-IR spectra are not needed when the table resumes all the results for it.

Answer: We thank the reviewer for the insightful comments and suggestions. We removed the FT-IR spectra from the manuscript. Thank you very much.

---

## [Decision Letter · Decision Letter 1]

8 May 2024

Silver nanoparticle biosynthesis utilizing Ocimum kilimandscharicum leaf extract and assessment of its antibacterial activity against certain chosen bacteria

PONE-D-23-37831R1

Dear Dr. Ouandaogo,

We’re pleased to inform you that your manuscript has been judged scientifically suitable for publication and will be formally accepted for publication once it meets all outstanding technical requirements.

Kind regards,

Jorddy Neves Cruz

Academic Editor

PLOS ONE

Additional Editor Comments (optional):

Reviewers' comments:

Reviewer's Responses to Questions

**Comments to the Author**

1. If the authors have adequately addressed your comments raised in a previous round of review and you feel that this manuscript is now acceptable for publication, you may indicate that here to bypass the “Comments to the Author” section, enter your conflict of interest statement in the “Confidential to Editor” section, and submit your "Accept" recommendation.

Reviewer #1: All comments have been addressed

2. Is the manuscript technically sound, and do the data support the conclusions?

Reviewer #1: Yes

3. Has the statistical analysis been performed appropriately and rigorously? 

Reviewer #1: Yes

4. Have the authors made all data underlying the findings in their manuscript fully available?

Reviewer #1: Yes

5. Is the manuscript presented in an intelligible fashion and written in standard English?

Reviewer #1: Yes

6. Review Comments to the Author

Reviewer #1: Reviewer’s Constructive Feedback to the Authors:

The revision of the manuscript titled "Silver nanoparticle biosynthesis utilizing Ocimum kilimandscharicum leaf extract and assessment of its antibacterial activity against certain chosen bacteria" reflects a commendable effort by the authors to address all queries raised during the initial review process. The manuscript has undergone significant improvements in clarity, methodology, and data presentation. The authors have successfully elucidated the significance of their research objectives and provided comprehensive details regarding the biosynthesis of silver nanoparticles using Ocimum kilimandscharicum leaf extract. Methodological enhancements, including the incorporation of controls and standardization techniques, have strengthened the experimental approach. Moreover, the presentation of experimental data through well-organized tables and figures, along with statistical analyses, enhances the reliability of the reported results. Minor revisions, such as adjustments to referencing style and language refinement, have also been effectively addressed. Based on the thorough revisions made, I recommend acceptance of the manuscript for publication, pending final approval from the editorial board.

7. PLOS authors have the option to publish the peer review history of their article (what does this mean?). If published, this will include your full peer review and any attached files.

Reviewer #1: **Yes: **Dr. Shahina Akter

---

## [Editor Report · Acceptance letter]

17 May 2024

PONE-D-23-37831R1 

PLOS ONE

Dear Dr. Ouandaogo, 

I'm pleased to inform you that your manuscript has been deemed suitable for publication in PLOS ONE. Congratulations! Your manuscript is now being handed over to our production team.

Kind regards, 

on behalf of

Dr. Jorddy Neves Cruz 

Academic Editor

PLOS ONE